# STDR: Spatio-Temporal Decoupling for Real-Time Dynamic Scene Rendering

## Abstract

Although dynamic scene reconstruction has long been a fundamental challenge in 3D vision, the recent emergence of 3D Gaussian Splatting (3DGS) offers a promising direction by enabling high-quality, real-time rendering through explicit Gaussian primitives. However, existing 3DGS-based methods for dynamic reconstruction often suffer from *spatio-temporal incoherence* during initialization, where canonical Gaussians are constructed by aggregating observations from multiple frames without temporal distinction. This results in spatio-temporally entangled representations, making it difficult to model dynamic motion accurately. To overcome this limitation, we propose **STDR** (Spatio-Temporal Decoupling for Real-time rendering), a plug-and-play module that learns spatio-temporal probability distributions for each Gaussian. STDR introduces a spatio-temporal mask, a separated deformation field, and a consistency regularization to jointly disentangle spatial and temporal patterns. Extensive experiments demonstrate that incorporating our module into existing 3DGS-based dynamic scene reconstruction frameworks leads to notable improvements in both reconstruction quality and spatio-temporal consistency across synthetic and real-world benchmarks.

## 1 Introduction

Dynamic scene reconstruction aims to recover the geometry, appearance, and motion of real-world environments where objects or agents exhibit time-varying behaviors. This capability is essential for a broad range of applications, such as virtual and augmented reality (Stotko et al., 2019; Kim et al., 2019), autonomous driving (Yan et al., 2024; Li et al., 2024a; Peng et al., 2024; Duan & Yang, 2024), and robotic manipulation (Lu et al., 2024; Zhang et al., 2024b). However, creating high-quality, temporally consistent reconstructions of dynamic scenes remains challenging due to complex motion patterns, occlusions, and the inherent sparsity of observations.

Over the past few years, novel view synthesis techniques have achieved remarkable progress. Neural Radiance Fields (NeRF) (Mildenhall et al., 2020) demonstrated impressive results for static scenes by learning continuous volumetric representations, yet its dense sampling strategy leads to high computational costs and slow inference speeds. In contrast, 3D Gaussian Splatting (3DGS) (Kerbl et al., 2023) adopts explicit Gaussian primitives, enabling not only high-quality reconstruction but also real-time rendering performance. These advances have naturally extended to dynamic scenes, with numerous methods (Huang et al., 2024; Yang et al., 2023; Wan et al., 2024; Wu et al., 2024; Zhu et al., 2024) incorporating temporal modeling through deformation fields or time-conditioned representations.

Despite recent progress, we observe that existing 3DGS-based methods for dynamic scene reconstruction still suffer from a critical limitation that remains insufficiently addressed. These methods typically adopt a two-stage pipeline: they first build a canonical representation by aggregating observations from multiple time frames, and then learn deformation fields to capture scene dynamics. However, this aggregation often leads to "spatio-temporal incoherence", where the canonical representation mixes information from different temporal states. Such inconsistency complicates the learning of deformation fields and undermines the accuracy of motion reconstruction.

As illustrated in Fig 1, when initializing canonical Gaussians from multi-frame observations without temporal distinction, dynamic objects appear in multiple positions simultaneously, creating representations with overlapping temporal states. For example, a moving excavator arm is represented by

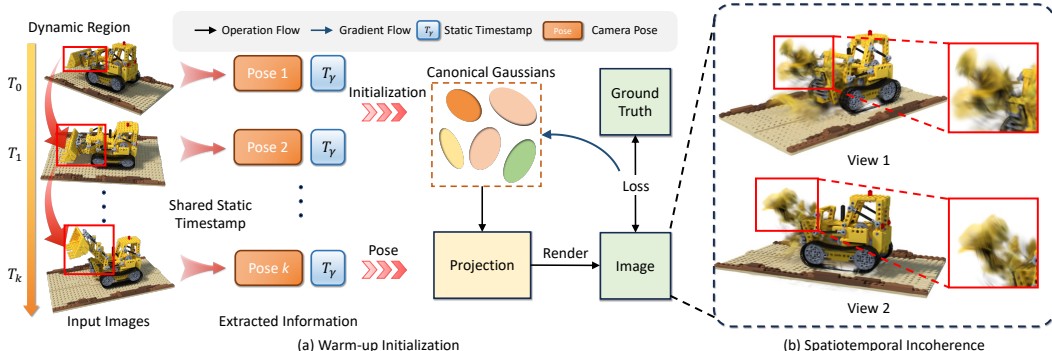

Figure 1: Visualization of spatio-temporal incoherence during canonical Gaussian initialization. (a) Initial Gaussians are generated by aggregating observations from different timestamps under a shared static timestamp, leading to temporally mixed representations. (b) Resulting ghosting artifacts reflect overlapping temporal states, which hinder the accurate motion patterns.

overlapping Gaussians at different points along its trajectory within the same canonical space. This initialization creates a problematic initial state: the deformation field must differentiate between Gaussians that are spatially close but temporally distant, without explicit temporal guidance.

This initialization-induced incoherence leads to significant challenges during deformation learning. The model faces inherent ambiguity when attempting to map temporally mixed Gaussians to their correct positions at specific timestamps. It must determine which Gaussians correspond to which time frames and resolve conflicts where the same spatial region needs to map to multiple different positions. Without addressing this ambiguity, reconstructions exhibit visible artifacts including ghosting effects, motion blur, and temporally inconsistent deformations.

Despite its impact on reconstruction quality, this fundamental problem has received limited attention in existing literature. While several methods (Zhu et al., 2024; Gao et al., 2024; Lin et al., 2025) have explored static-dynamic decomposition by separating backgrounds from moving objects, they primarily focus on spatial disentanglement without resolving the underlying temporal ambiguity that arises during initialization. This limitation persists across different architectural choices and representation techniques.

To address this critical issue, we propose **STDR** (Spatio-Temporal Decoupling for Real-time rendering), a general and plug-and-play module that can be seamlessly integrated into existing 3DGS-based pipelines. STDR explicitly disentangles the spatio-temporal relationships of Gaussians by learning their probability distributions across space and time. It comprises three key components: (1) a spatio-temporal mask that modulates opacity to capture temporal activation patterns, (2) a separated deformation field that leverages spatio-temporal features to factorize temporal and spatial structure, and (3) spatio-temporal consistency regularization that enforces smooth and coherent scene dynamics.

By integrating these components, STDR enables the model to distinguish between Gaussians that are spatially close but temporally distant, improving motion disentanglement and static-dynamic decomposition. During training, the spatio-temporal masks are optimized via backpropagation and gradually converge to temporal activation distributions that reflect the true dynamics of the scene. The separated deformation field further guides each Gaussian toward temporally aligned positions using factorized spatial and temporal embeddings. Finally, temporal smoothness and spatial-awareness regularizations encourage coherent transitions over time and similarity among spatial neighbors, reinforcing both temporal alignment and geometric stability.

We comprehensively evaluate the effectiveness of STDR on three widely-used dynamic scene datasets, namely D-NeRF Pumarola et al. (2020), NeRF-DS(Yan et al., 2023) and HyperNeRF (Park et al., 2021b). To ensure a fair and rigorous comparison, we incorporate STDR into four representative baseline methods: 4DGS (Wu et al., 2024), DeformGS (Yang et al., 2023), SPGS (Wan et al., 2024) and SC-GS (Huang et al., 2024), and conduct controlled experiments based on their original implementations. Quantitative results consistently demonstrate that our method leads to substantial improvements in reconstruction quality, as reflected by increased PSNR and SSIM scores and decreased LPIPS values. These gains highlight the ability of STDR to enhance both the perceptual

fidelity and structural accuracy of dynamic scene reconstruction, while effectively mitigating the spatio-temporal incoherence problem discussed earlier.

**In summary, our main contributions are:**

- We analyze the problem of "spatio-temporal incoherence" in dynamic scene reconstruction, which arises from temporally mixed initialization of Gaussians.

- We propose **STDR**, a plug-and-play dual spatio-temporal decoupling module that disentangles temporal and spatial patterns via spatio-temporal masks and separated deformation fields.

- We introduce a spatio-temporal consistency regularization that enforces smooth temporal evolution and spatial structural awareness to stabilize the deformation process.

- Extensive experiments demonstrate that STDR significantly improves reconstruction quality across dynamic scene benchmarks, effectively addressing the initialization-induced incoherence.

## 2 PRELIMINARY

### 2.1 3D GAUSSIAN SPLATTING

3D Gaussian Splatting (3DGS) (Kerbl et al., 2023) is a novel and efficient technique that represents and renders 3D scenes using a set of 3D Gaussians. Each 3D Gaussian is characterized by a center point $\mathcal{X}$, representing the mean of the distribution, and a covariance matrix $\Sigma$, capturing its spatial extent and orientation. The corresponding contribution of a 3D Gaussian can be formulated as:

$$G(X) = e^{-\frac{1}{2}(X-\mathcal{X})^T \Sigma^{-1}(X-\mathcal{X})} \tag{1}$$

To simplify the learning process of 3D Gaussians, the $\Sigma$ can be decoupled into two components: the rotation matrix $R$, and the scaling matrix $S$:

$$\Sigma = RSS^T R^T \tag{2}$$

Then each Gaussian can be projected onto the 2D camera plane and renders the influence of the Gaussian on each pixel using its corresponding 2D covariance matrix $\Sigma^{'} = JV\Sigma V^T J^T$, where $J$ refers to the Jacobian of the affine approximation of the projective transformation, and $V$ represents the view matrix that maps coordinates from world space to camera space. The color $C$ of the pixel on image plane is computed through $\alpha$-blending with depth-ordered $N$ Gaussians overlapped the pixel:

$$C = \sum_{i \in \mathcal{N}} c_i \alpha_i \prod_{j=1}^{i-1}(1 - \alpha_j), \tag{3}$$

where $c_i, \alpha_i$ are the color and the blending weight of the i-th 3D Gaussian.

### 2.2 DYNAMIC SCENE RECONSTRUCTION WITH DEFORMATION FIELDS

Dynamic Neural Radiance Field (NeRF) methods extend the original NeRF framework (Mildenhall et al., 2020) to handle time-varying scenes. These approaches typically extend the radiance field with a temporal component, allowing the scene representation to dynamically evolve over time.

Inspired by dynamic NeRF-based methods, recent 3D Gaussian-based approaches integrate temporal deformation fields to model dynamic scenes. A deformation function $f_{\text{def}}(x, t)$ takes the original Gaussian attributes, which include position $x$, rotation $r$, scale $s$, color $c$, and opacity $\alpha$, and produces a residual update $\Delta G = (\delta x, \delta r, \delta s, \delta c, \delta \alpha)$ at a given time $t$. This update represents the time-dependent changes in the Gaussian parameters. The updated Gaussian $G_t$ is then obtained by applying the deformation to the original attributes:

$$G_t = \{x, r, s, c, \alpha\} + f_{\text{def}}(x, t) = \{x + \delta x, \ r + \delta r, \ s + \delta s, \ c + \delta c, \ \alpha + \delta \alpha\}. \tag{4}$$

While this approach enables modeling of dynamic scenes, the deformation fields typically operate on canonical Gaussians without considering their spatio-temporal coherence during initialization. This limitation often leads to ambiguous motion patterns and inconsistent reconstructions, particularly in scenes with complex dynamics.

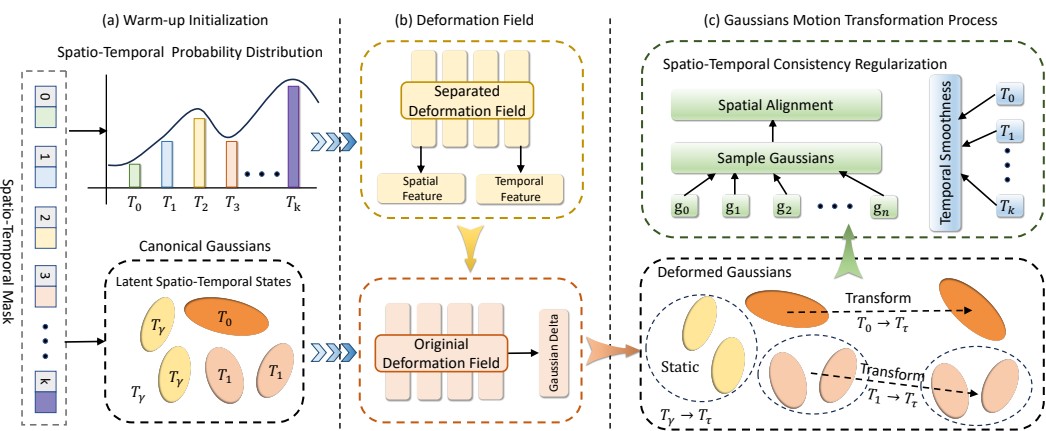

Figure 2: Overview of the proposed method. (a) During warm-up initialization, we assign each Gaussian a learnable spatio-temporal mask to capture its spatial and temporal correlations, which is further refined into a spatio-temporal probability distribution. (b) A separated deformation field decouples spatial and temporal features, enabling factorized modeling of motion and structure. (c) During deformation, each Gaussian infers its temporal identity and motion type, enabling accurate alignment with its target timestamp. Spatio-temporal consistency regularization further enhances this process by promoting smooth transitions and coherent structural alignment.

## 3 METHOD

### 3.1 OVERVIEW

Our goal is to enable high-quality dynamic scene reconstruction while addressing the spatio-temporal incoherence introduced during initialization. We propose **STDR**, a general plug-and-play module that can be seamlessly integrated into existing 3D Gaussian-based frameworks.

As shown in Figure 2, STDR explicitly learns and decouples the spatio-temporal relationships of Gaussians via three core components: (1) a spatio-temporal mask that encodes probability distributions over time, (2) a separated deformation field that factorizes spatial structure and temporal dynamics, and (3) a consistency regularization that encourages coherent motion and spatial alignment.

In the following sections, we first highlight the phenomenon of "spatio-temporal incoherence" in existing methods (Section 3.2). We then present the design of our STDR module in Section 3.3, followed by a detailed explanation of its spatio-temporal consistency regularization in Section 3.4.

### 3.2 PHENOMENON OF SPATIOTEMPORAL INCOHERENCE

Most existing 3DGS-based methods follow a two-stage pipeline: initializing canonical Gaussians from all input frames, then learning deformation fields to model dynamic scene. This seemingly straightforward approach, however, creates a fundamental challenge we term "spatio-temporal incoherence".

As illustrated in Figure 1 (a), during the warm-up initialization phase, all input frames are processed as if captured at a shared timestamp $t_\gamma$. The resulting canonical Gaussians inevitably capture multiple temporal states of dynamic objects superimposed together. For instance, a moving excavator's arm gets represented by overlapping Gaussians at different positions along its motion trajectory, creating a "ghosted" appearance in the canonical space (as illustrated in Figure 1 (b)).

This initialization leads to severe ambiguity when learning deformation fields. As shown in Figure 2 (b), the original deformation field must somehow map these spatially entangled Gaussians to their correct temporal instances.

However, without explicit temporal information, the deformation field faces two critical challenges: First, it cannot determine which Gaussians belong to which timestamp, multiple Gaussians may occupy similar positions but represent different temporal states. Second, the spatial overlap between

Gaussians from different timestamps creates conflicting deformation targets, where the same spatial region needs to be mapped to multiple different positions depending on the timestamp.

These ambiguities manifest as artifacts in the reconstruction: ghosting effects where objects appear semi-transparent, motion blur along trajectories, and temporally inconsistent deformations where parts of objects move unnaturally. The root cause is that traditional deformation fields operate solely on positional features without understanding the underlying spatio-temporal structure. They lack the mechanism to distinguish between Gaussians that are spatially close but temporally distant, leading to the incoherent motion patterns observed in practice.

## 3.3 SPATIO-TEMPORAL DECOUPLING MODULE

To address these challenges, we introduce a spatio-temporal decoupling module that explicitly learns and separates the temporal and spatial characteristics of each Gaussian.

**Spatio-Temporal Mask** While previous methods have utilized dynamic-static masks to differentiate between dynamic and static regions in a scene, we are the first to introduce a **spatio-temporal mask** that jointly captures both temporal and spatial characteristics. We augment each Gaussian with a learnable temporal mask $\mathbf{m}_i \in \mathbb{R}^K$, where $K$ is the number of time frames. This mask captures the probability distribution of Gaussian i across all timestamps. During rendering, we modulate the original opacity $\alpha_i$ with the temporal mask:

$$\alpha_i^{\text{new}} = m_i^j \cdot \alpha_i, \tag{5}$$

where $j$ denotes the current timestamp. Through training, these masks learn to activate Gaussians at appropriate timestamps, effectively resolving temporal ambiguity.

**Spatio-Temporal Probability Distribution** During training with images captured at varying timestamps and camera poses, spatio-temporal mask modulated opacity $\alpha_i^{\text{new}}$ of each Gaussian is optimized via backpropagation, progressively refining its temporal mask $m_i$. Gaussians located in static regions are inherently insensitive to temporal variations and thus receive relatively consistent gradients across different time steps.

In contrast, Gaussians associated with dynamic regions exhibit larger rendering errors when supervised with images from mismatched temporal frames. This temporal misalignment leads to stronger backpropagation gradients, which in turn drive more substantial updates, enabling these Gaussians to better adapt to their correct temporal representations. Finally, we normalize each Gaussian's mask along the temporal dimension to obtain its spatio-temporal probability distribution $\tilde{m}_i^j$:

$$\tilde{m}_i^j = \frac{\exp(m_i^j)}{\sum_{k=1}^{K} \exp(m_i^k)} \tag{6}$$

**Separated Deformation Field** Instead of directly deforming Gaussians based on position alone, we introduce a separated deformation field that leverages the learned spatio-temporal probability distributions:

$$z_s, z_t = f_{\text{sep}}(x_\gamma, \tilde{m}_\gamma)$$
$$G_\tau = f_{\text{def}}(x_\gamma, z_s, z_t, t_\tau) \tag{7}$$

where $f_{\text{sep}}$ extracts spatial features $z_s$ and temporal features $z_t$ from the Gaussian position $x_\gamma$ and its probability distribution $\tilde{m}_\gamma$.

This separation enables more accurate deformation in several ways. The spatial features $z_s$ capture the structural context of each Gaussian, allowing the deformation field to distinguish between static background and dynamic objects. Meanwhile, the temporal feature $z_t$ encodes time-dependent motion patterns, enabling the assignment of temporal identities to Gaussians. Gaussians with the same temporal identity exhibit consistent motion behaviors, as illustrated in Fig. 2 (c).

## 3.4 SPATIO-TEMPORAL CONSISTENCY REGULARIZATION

To further enhance the coherence of our reconstructions, we introduce two regularization terms that encourage spatio-temporal consistency:

Table 1: Quantitative comparisons on the D-NeRF dataset (Pumarola et al., 2020). Evaluated on full-resolution (800×800) images using PSNR, SSIM, and LPIPS (VGG). STDR, integrated into DeformGS (Yang et al., 2023) and SC-GS (Huang et al., 2024), consistently improves performance, demonstrating the effectiveness of spatio-temporal decoupling. We highlight the improvements achieved by incorporating STDR.

| Method | Hell Warrior | | | Mutant | | | Hook | | | Bouncing Balls | | |
|---|---|---|---|---|---|---|---|---|---|---|---|---|
| | PSNR↑ | SSIM↑ | LPIPS↓ | PSNR↑ | SSIM↑ | LPIPS↓ | PSNR↑ | SSIM↑ | LPIPS↓ | PSNR↑ | SSIM↑ | LPIPS↓ |
| 3D-GS (Kerbl et al., 2023) | 29.89 | 0.916 | 0.106 | 24.53 | 0.934 | 0.058 | 21.71 | 0.888 | 0.103 | 23.20 | 0.959 | 0.060 |
| D-NeRF (Pumarola et al., 2020) | 24.06 | 0.944 | 0.071 | 30.31 | 0.967 | 0.039 | 29.02 | 0.960 | 0.055 | 38.17 | 0.989 | 0.032 |
| TiNeuVox (Fang et al., 2022) | 27.10 | 0.964 | 0.077 | 31.87 | 0.961 | 0.047 | 30.61 | 0.960 | 0.059 | 40.23 | 0.993 | 0.042 |
| Tensor4D (Shao et al., 2023) | 31.26 | 0.925 | 0.074 | 29.11 | 0.945 | 0.060 | 28.63 | 0.943 | 0.064 | 24.47 | 0.962 | 0.044 |
| K-Planes (Fridovich-Keil et al., 2023) | 24.58 | 0.952 | 0.082 | 32.50 | 0.971 | 0.036 | 28.12 | 0.949 | 0.066 | 40.05 | 0.993 | 0.032 |
| Deformable3D (Yang et al., 2023) | 41.54 | 0.987 | 0.023 | 42.63 | 0.995 | 0.005 | 37.42 | 0.987 | 0.014 | 41.01 | 0.995 | 0.009 |
| +STDR | 42.22 | 0.989 | 0.021 | 42.84 | 0.995 | 0.005 | 38.17 | 0.989 | 0.012 | 41.53 | 0.995 | 0.009 |
| SCGS (Huang et al., 2024) | 42.19 | 0.989 | 0.019 | 43.43 | 0.996 | 0.005 | 38.79 | 0.990 | 0.009 | 41.59 | 0.995 | 0.009 |
| +STDR | **42.45** | **0.993** | **0.014** | **43.66** | **0.999** | **0.002** | **39.43** | **0.997** | **0.007** | **42.45** | **0.997** | **0.004** |

| Method | T-Rex | | | Stand Up | | | Jumping Jacks | | | Mean | | |
|---|---|---|---|---|---|---|---|---|---|---|---|---|
| | PSNR↑ | SSIM↑ | LPIPS↓ | PSNR↑ | SSIM↑ | LPIPS↓ | PSNR↑ | SSIM↑ | LPIPS↓ | PSNR↑ | SSIM↑ | LPIPS↓ |
| 3D-GS (Kerbl et al., 2023) | 21.93 | 0.954 | 0.049 | 21.91 | 0.930 | 0.079 | 20.64 | 0.930 | 0.083 | 23.40 | 0.930 | 0.077 |
| D-NeRF (Pumarola et al., 2020) | 30.61 | 0.967 | 0.054 | 33.13 | 0.978 | 0.036 | 32.70 | 0.978 | 0.039 | 31.14 | 0.969 | 0.047 |
| TiNeuVox (Fang et al., 2022) | 31.25 | 0.967 | 0.048 | 34.61 | 0.980 | 0.033 | 33.49 | 0.977 | 0.041 | 32.74 | 0.972 | 0.050 |
| Tensor4D (Shao et al., 2023) | 23.86 | 0.935 | 0.054 | 30.56 | 0.958 | 0.036 | 24.20 | 0.925 | 0.067 | 27.44 | 0.942 | 0.057 |
| K-Planes (Fridovich-Keil et al., 2023) | 30.43 | 0.974 | 0.034 | 33.10 | 0.979 | 0.031 | 31.11 | 0.971 | 0.047 | 31.41 | 0.970 | 0.047 |
| Deformable3D (Yang et al., 2023) | 38.10 | 0.993 | 0.010 | 44.62 | 0.995 | 0.006 | 37.72 | 0.990 | 0.013 | 40.43 | 0.992 | 0.011 |
| +STDR | 38.76 | 0.994 | 0.009 | 44.94 | 0.995 | 0.006 | 38.52 | 0.991 | 0.011 | 41.00 | 0.993 | 0.010 |
| SCGS (Huang et al., 2024) | 39.53 | 0.994 | 0.009 | 46.72 | 0.997 | 0.004 | 39.34 | 0.992 | 0.008 | 41.66 | 0.993 | 0.009 |
| +STDR | **40.45** | **0.999** | **0.005** | **46.88** | **0.999** | **0.003** | **40.35** | **0.997** | **0.006** | **42.24** | **0.997** | **0.006** |

**Temporal Smoothness Regularization**   We encourage spatio-temporal masks to change smoothly across adjacent timestamps:

$$\mathcal{L}_{\text{temp}} = \lambda_1 \sum_i \sum_t \left\| m_i^t - m_i^{t+1} \right\|_2^2 , \tag{8}$$

This prevents abrupt temporal transitions and promotes natural motion patterns.

**Spatial-Awareness Regularization**   We encourage spatially adjacent Gaussians to share similar temporal behaviors:

$$\mathcal{L}_{\text{spatial}} = \lambda_2 \frac{1}{MK} \sum_{i=1}^{M} \sum_{j=1}^{K} \sum_{d=1}^{D} m_i^d \log\left(\frac{m_i^d}{m_j^d}\right) , \tag{9}$$

where $M$ denotes the number of sampled Gaussians, $K$ denotes the number of the neighbor Gaussians, and $D$ denotes the number of all timestamps.

The final loss combines these regularizations with the standard reconstruction loss:

$$\mathcal{L} = \lambda \mathcal{L}_{\text{recon}} + \lambda_1 \mathcal{L}_{\text{temp}} + \lambda_2 \mathcal{L}_{\text{spatial}} = \lambda \mathcal{L}_1 + (1-\lambda)\mathcal{L}_{\text{D-SSIM}} + \lambda_1 \mathcal{L}_{\text{temp}} + \lambda_2 \mathcal{L}_{\text{spatial}} \tag{10}$$

## 4 EXPERIMENT

### 4.1 EXPERIMENTAL SETTINGS

**Implementation Details**   Our implementation is tested on a single A100 GPU. Following the same setting as for opacity, we learn the spatio-temporal mask by applying sigmoid activation function. We normalize the temporal dimension using softmax function to obtain a probability distribution over time before feeding it into the deformation field. We employ a 6-layer MLP as the separated deformation field. And we evaluate our experimental results using image quality metrics, including Peak Signal-to-Noise Ratio (PSNR), Structural Similarity Index (SSIM (Wang et al., 2004)), and Learned Perceptual Image Patch Similarity (LPIPS) (Zhang et al., 2018).

**Datasets**   To validate the effectiveness of our method, we conduct extensive experiments on both synthetic and real-world datasets, including the D-NeRF (Pumarola et al., 2020), NeRF-DS (Yan et al., 2023), and HyperNeRF (Park et al., 2021b). The D-NeRF dataset contains eight scenes featuring various articulated and non-rigid deformations, along with ground-truth geometry and

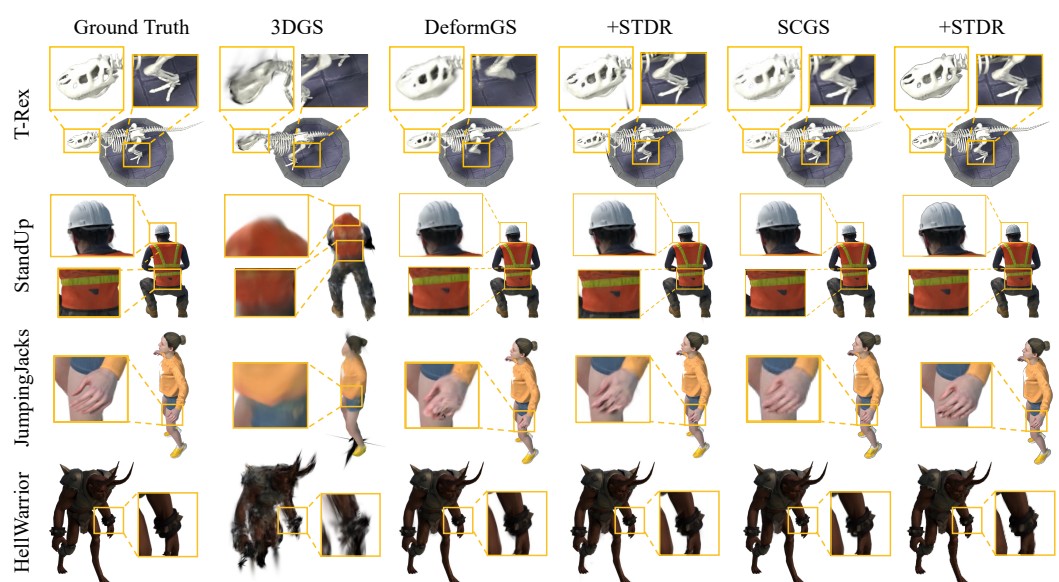

Figure 3: Visualization of Comparisons on D-NeRF Dataset (Pumarola et al., 2020).

calibrated camera poses for each frame. For real-world evaluation, we adopt the NeRF-DS and HyperNeRF datasets. NeRF-DS includes real monocular video sequences with annotated camera trajectories and moderate object motion. HyperNeRF captures complex real-life deformations and topological changes using one or two moving cameras.

**Baselines** We evaluate our method against several approaches that utilize Gaussian representations for dynamic scene reconstruction. Among them, 4D-GS (Wu et al., 2024) and DeformGS (Yang et al., 2023) adopt distinct paradigms: 4D-GS employs a planar and explicit representation, whereas DeformGS utilizes a fully implicit MLP-based deformation field. SC-GS (Huang et al., 2024) and SPGS (Wan et al., 2024) are both extensions of DeformGS.

## 4.2 EXPERIMENTAL COMPARISONS

**Comparative Analysis of Synthetic Datasets** In our experiments, we integrate the proposed STDR module into DeformGS and SC-GS, and perform quantitative comparisons with prior methods on the D-NeRF dataset. Notably, the Lego scene in the D-NeRF dataset exhibits a distribution shift between the training and testing sets. To ensure fair and meaningful evaluation, we exclude this scene and report results on the remaining seven scenes. As shown in Table 1, incorporating STDR results in significant performance improvements across all metrics.

To further demonstrate the effectiveness of our method, we present qualitative results in Figure 3. STDR enables clearer modeling of dynamic content and more consistent structural representation, thereby improving overall reconstruction quality. For instance, in the T-Rex scene, the baseline model struggles to accurately capture the foot due to spatio-temporal incoherence during initialization, resulting in ghosting and structural ambiguity. With STDR, the foot region is reconstructed with greater temporal consistency and geometric fidelity.

**Comparative Analysis of Real-world Datasets** Beyond experiments on the synthetic dataset, we further integrate the proposed STDR module into representative baselines and evaluate its effectiveness on two real-world datasets: NeRF-DS (Yan et al., 2023) and HyperNeRF (Park et al., 2021b). As shown in Table 2 and Table 4, incorporating STDR leads to consistent and significant improvements in reconstruction quality across diverse dynamic scenes, with notable gains in both PSNR and SSIM metrics.

Figure 4 presents qualitative results on the NeRF-DS dataset, where STDR improves the temporal consistency of SPGS reconstructions, particularly in regions with fast motion or complex deformations. Similarly, as shown in Figure 5, STDR effectively mitigates severe artifacts observed in the broom scene of HyperNeRF when using 4D-GS, enhancing overall reconstruction fidelity.

Table 2: Quantitative comparison on the NeRF-DS dataset (Pumarola et al., 2020). STDR, integrated into SPGS (Wan et al., 2024), consistently improves reconstruction quality across diverse real-world scenes. We highlight the improvements achieved by incorporating STDR.

| Method | Sieve | | | Plate | | | Bell | | | Press | | |
|---|---|---|---|---|---|---|---|---|---|---|---|---|
| | PSNR↑ | SSIM↑ | LPIPS↓ | PSNR↑ | SSIM↑ | LPIPS↓ | PSNR↑ | SSIM↑ | LPIPS↓ | PSNR↑ | SSIM↑ | LPIPS↓ |
| 3D-GS Kerbl et al. (2023) | 23.16 | 0.8203 | 0.2247 | 16.14 | 0.6970 | 0.4093 | 21.01 | 0.7885 | 0.2503 | 22.89 | 0.8163 | 0.2904 |
| TiNeuVox Fang et al. (2022) | 21.49 | 0.8265 | 0.3176 | **20.58** | **0.8027** | 0.3317 | 23.08 | 0.8242 | 0.2568 | 24.47 | 0.8613 | 0.3001 |
| SPGS (Wan et al., 2024) | 25.20 | 0.8640 | 0.1584 | 19.04 | 0.7734 | 0.2662 | 25.16 | 0.8433 | 0.1702 | 23.77 | 0.8408 | 0.2662 |
| +STDR | **25.70** | **0.8689** | **0.1562** | 19.41 | 0.7808 | 0.2641 | **25.30** | **0.8451** | **0.1695** | 24.64 | 0.8467 | 0.2581 |

| Method | Cup | | | As | | | Basin | | | Mean | | |
|---|---|---|---|---|---|---|---|---|---|---|---|---|
| | PSNR↑ | SSIM↑ | LPIPS↓ | PSNR↑ | SSIM↑ | LPIPS↓ | PSNR↑ | SSIM↑ | LPIPS↓ | PSNR↑ | SSIM↑ | LPIPS↓ |
| 3D-GS Kerbl et al. (2023) | 21.71 | 0.8304 | 0.2548 | 22.69 | 0.8017 | 0.2994 | 18.42 | 0.7170 | 0.3153 | 20.86 | 0.7816 | 0.2920 |
| TiNeuVox Fang et al. (2022) | 19.71 | 0.8109 | 0.3643 | 21.26 | 0.8289 | 0.3967 | **20.66** | **0.8145** | 0.2690 | 21.61 | 0.8241 | 0.3195 |
| SPGS (Wan et al., 2024) | 24.26 | 0.8810 | 0.1747 | 24.70 | 0.8638 | 0.2196 | 19.23 | 0.7727 | 0.2087 | 23.05 | 0.8341 | 0.2091 |
| +STDR | **24.36** | **0.8817** | **0.1743** | **25.23** | **0.8703** | **0.2055** | 19.44 | 0.7824 | 0.2024 | **23.44** | **0.8394** | **0.2043** |

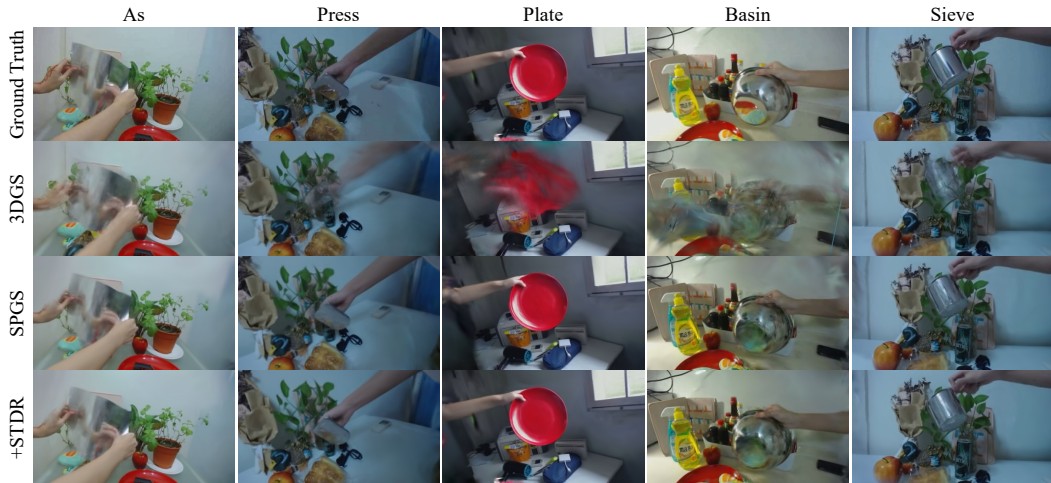

Figure 4: Visualization of Comparisons on NeRF-DS Dataset (Yan et al., 2023).

## 4.3 ABLATION STUDIES

In Table 3, we conduct ablation studies on the proposed spatio-temporal consistency regularization. Our method is integrated into SC-GS and evaluated on the D-NeRF dataset, with results averaged across all seven test scenes. The removal of either the temporal smoothness loss $\mathcal{L}_{\text{temp}}$ or the spatial-awareness loss $\mathcal{L}_{\text{spatial}}$ results in a noticeable drop in reconstruction quality, highlighting the critical role of both components in achieving stable and coherent dynamic reconstructions.

Table 3: Ablation studies on spatio-temporal consistency regularization using the D-NeRF dataset.

| Method | PSNR↑ | SSIM↑ | LPIPS↓ |
|---|---|---|---|
| SC-GS | 41.66 | 0.993 | 0.0090 |
| +STDR | **42.24** | **0.997** | **0.0059** |
| w/o $\mathcal{L}_{\text{temp}}$ | 41.91 | 0.996 | 0.0067 |
| w/o $\mathcal{L}_{\text{spatial}}$ | 42.11 | 0.997 | 0.0063 |

## 5 RELATED WORKS

### 5.1 SCENE REPRESENTATION FOR 3D RECONSTRUCTION

Recent advances in 3D scene representation have significantly improved the quality and efficiency of 3D reconstruction. Early works (Newcombe et al., 2015; Qi et al., 2017) predominantly rely on mesh-based or point-based geometry to capture spatial structures, but such methods often struggle to handle complex appearance or lighting effects. Implicit representations, most notably Neural Radiance Fields (NeRF) (Mildenhall et al., 2020), model scenes as continuous volumetric functions using neural networks, enabling photorealistic rendering from sparse inputs. Building upon this, various extensions (Barron et al., 2021; Hu et al., 2022; Arandjelović & Zisserman, 2021; Sun et al., 2023) have improved rendering quality, sampling strategies, and training efficiency. How-

ever, NeRF-based approaches still require dense sampling and costly optimization, limiting their applicability in real-time or large-scale settings.

To overcome these limitations, 3D Gaussian Splatting (3DGS) (Kerbl et al., 2023) has emerged as an explicit representation that models scenes with a set of 3D Gaussian ellipsoids. This structure combines the benefits of neural rendering and explicit geometry, enabling fast, high-fidelity visualization. Recent works have extended 3DGS to semantic segmentation (Wang et al., 2024a; Shen et al., 2024; Choi et al., 2024; Ye et al., 2024), boundary modeling (Qu et al., 2024; Li et al., 2024c) and large-scale environments (Lin et al., 2024; Ren et al., 2024; Liu et al., 2024; Fan et al., 2024; Zhang et al., 2024a), while also exploring real-time acceleration (Radl et al., 2024; Li et al., 2024b; Hanson et al., 2025; Wang et al., 2025a; 2024b).

### 5.2 DYNAMIC SCENE RECONSTRUCTION

**NeRF-based Dynamic Scene Reconstruction**   Initially, D-NeRF (Pumarola et al., 2020) extends NeRF by incorporating time as an additional input and learning a deformation field to model object motions across frames, enabling the reconstruction of simple non-rigid scenes. Building upon this idea, subsequent methods such as NSFF (Li et al., 2021) and HyperNeRF (Park et al., 2021b) introduce more sophisticated scene flow or higher-dimensional latent spaces to represent complex topology changes and motion discontinuities. To improve the robustness of motion estimation, approaches like Nerfies (Park et al., 2021a) and NerfPlayer (Song et al., 2023) propose optimization strategies based on tracking or factorized representations. Recently, advances such as DyNeRF (Li et al., 2022) leverage space-time video inputs and global scene priors to boost the temporal consistency and fidelity of dynamic reconstructions. While these NeRF-based dynamic models show promising results, most of them suffer from slow training and inference speeds due to their fully implicit representations, which limits their scalability and real-time applicability.

**3DGS-based Dynamic Scene Reconstruction**   Leveraging the real-time rendering advantages of 3DGS, several recent works have extended this framework to dynamic scene reconstruction. 4DGS (Wu et al., 2024) integrates dynamic modeling by learning per-frame transformations of Gaussians, enabling high-fidelity rendering of deforming scenes. DeformGS (Yang et al., 2023) introduces deformation fields that warp canonical Gaussians into target frames, facilitating scene dynamics through continuous motion modeling. SC-GS (Huang et al., 2024) further enhances spatial consistency by proposing structural constraints to regulate the Gaussian distribution.

While numerous methods (Zhu et al., 2024; Gao et al., 2024; Lin et al., 2025) have been proposed to address static-dynamic decomposition by separating static backgrounds from moving objects within the spatial domain, these methods improve the spatial alignment of Gaussians over time but predominantly focus on spatial disentanglement. They often overlook the entangled temporal behaviors that arise from inconsistent initialization. Recent dynamic 4D Gaussian methods such as SpacetimeGS (Li et al., 2024d) and FreeTimeGS (Wang et al., 2025b) further enhance temporal modeling by extending Gaussian primitives to 4D representations in $(x, y, z, t)$, for example through spatio-temporal feature splatting or time-dependent visibility and opacity modeling. Unlike our approach, which is designed as a reusable spatio-temporal prior that can be plugged into different canonical–deformation pipelines.

In parallel, studies from other domains (Cai et al., 2019; 2020) have explored spatio-temporal modeling strategies, such as graph-based and propagation-based techniques. Although these approaches differ from our Gaussian-based framework in both application scope and technical formulation, their emphasis on spatio-temporal reasoning provides conceptual insights relevant to our work. In contrast, our proposed method introduces a novel dual spatio-temporal decoupling framework that disentangles both spatial structure and temporal variation.

## 6 CONCLUSION

In this work, we identify "spatio-temporal incoherence" during initialization as a key bottleneck in dynamic scene reconstruction with 3D Gaussian Splatting. This issue arises when canonical Gaussians are constructed from multi-frame observations without temporal distinction, resulting in ghosting artifacts and ambiguous deformation targets. To address this, we propose STDR, a plug-

and-play spatio-temporal decoupling module that introduces spatio-temporal masks, a separated deformation field, and consistency regularization to explicitly disentangle spatial structure and temporal relationships. By integrating STDR into various Gaussian-based reconstruction pipelines, we consistently achieve substantial improvements in reconstruction fidelity, temporal alignment, and structural coherence across synthetic and real-world dynamic scene benchmarks.

## ETHICS STATEMENT

This work does not raise any ethical concerns. It does not involve human subjects, personally identifiable information, or sensitive data. No potentially harmful insights, methodologies, or applications are introduced. The datasets and models used are publicly available and widely adopted in prior research. We have complied with all relevant ethical standards and the ICLR Code of Ethics.

## REPRODUCIBILITY STATEMENT

We have made significant efforts to ensure the reproducibility of our work. The paper provides detailed descriptions of the proposed methodology in Section 3. Complete model architecture, training settings, experimental protocols, hyperparameters, and evaluation metrics are documented in Section 4 and Appendix A. All datasets used are publicly available and the preprocessing steps are described in the appendix.

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

# A  APPENDIX

In this supplementary material, we provide more specific details of our method. In Section A.1, we present more experimental details. In Section A.2, we provide additional experimental results. In Section A.3, we present analyses and discussions to clarify any confusing or unclear part of our method.

## A.1  MORE IMPLEMENTATION DETAILS

**More Implementation Details**  For each Gaussian, we introduce a learnable spatio-temporal mask that modulates its opacity, effectively replacing the original opacity to capture its latent spatio-temporal identity. The spatio-temporal mask is represented as a $K$-dimensional vector with a total size of $N \times K$, where $N$ denotes the number of Gaussians in the spatial domain, and $K$ denotes the number of input timestamps.

Specifically, during the initialization and warm-up phases, we disable the gradient flow to the original opacity values and only allow gradients to be backpropagated through the spatio-temporal mask. This ensures that the temporal activation patterns are learned independently, without interference from the original opacity. After the mask converges to a stable spatio-temporal distribution, we resume the optimization of the original opacity, allowing it to be fine-tuned based on the learned temporal semantics. This design decouples temporal identity learning from static appearance modeling, enabling better separation of dynamic behaviors and more precise temporal alignment.

During training, we set the warm-up initialization phase to the first 3000 iterations. In this stage, the model optimizes the spatio-temporal mask independently, while the original opacity remains frozen. From iteration 0 to 6000, we activate the spatio-temporal consistency regularization to guide the mask toward smooth and coherent spatio-temporal distributions. After 6000 iterations, the learned mask parameters are frozen and normalized using the softmax function across the temporal dimension, forming a spatio-temporal probability distribution. This distribution is then passed into the subsequent separated deformation field to encode each Gaussian's temporal and spatial features, enabling accurate motion modeling and structure-aware deformation.

We design a lightweight multi-branch network, separated deformation field, to extract the temporal identity and motion attribute of each Gaussian from its spatio-temporal mask. The network takes the spatio-temporal mask vector as input and first processes it through two shared fully connected layers to extract intermediate features. These features are then fed into two branches: the first branch predicts a continuous temporal feature vector using a two-layer MLP with a final `tanh` activation, capturing the temporal identity of the Gaussian; the second branch predicts a binary probability via a `sigmoid` function to classify whether the Gaussian is dynamic or static. Both branches incorporate Batch Normalization and Dropout layers to enhance generalization. This module provides auxiliary cues for downstream spatio-temporal modeling while maintaining scalability and robustness.

The entire training process is conducted using the Adam optimizer on an A100 GPU. The detailed parameter settings are as follows: $\lambda_1 = 0.1$, $\lambda_2 = 0.2$, with the number $N$ of local neighbors set to 5 for KNN. Additionally, the number $M$ of target Gaussians sampled for calculating the KL divergence is set to 1000, with a maximum sampling cap of 20000 to ensure computational efficiency.

**More experiment Details**  For synthetic data experiments, we adopt the D-NeRF dataset (Pumarola et al., 2020), which includes diverse scenes with non-rigid deformations. However, we exclude the Lego scene from our evaluation. It is worth noting that the Lego scene exhibits a clear discrepancy between the training and test sets, as evidenced by the significantly different flip angles of the Lego shovel—a concern also acknowledged in the DeformGS paper (Yang et al., 2023). All experiments on D-NeRF are conducted at a resolution of 800×800 pixels to preserve high-frequency details.

For real-world evaluations, we select two dynamic datasets: NeRF-DS (Yan et al., 2023) and Hyper-NeRF (Park et al., 2021b). On NeRF-DS, we follow the resolution setting of 480×270 pixels, which is suitable for its challenging scenes containing specular reflections and reflective surfaces, offering a rigorous test of reconstruction robustness. For HyperNeRF, we evaluate our method on five representative scenes: broom, chicken, cut lemon, peel banana, and printer. These scenes cover a variety

Table 4: Quantitative Comparisons on HyperNeRF Dataset (Park et al., 2021b). STDR, integrated into 4DGS (Wu et al., 2024), consistently improves reconstruction quality across diverse real-world scenes.

| Method | Broom | | Chicken | | Cut Lemon | | Peel Banana | | Printer | | Mean | |
|---|---|---|---|---|---|---|---|---|---|---|---|---|
| | PSNR | SSIM | PSNR | SSIM | PSNR | SSIM | PSNR | SSIM | PSNR | SSIM | PSNR | SSIM |
| HyperNeRF Park et al. (2021b) | 19.5 | 0.21 | 27.4 | 0.63 | 31.8 | 0.96 | 22.1 | 0.72 | 20.0 | 0.63 | 24.16 | 0.63 |
| TiNeuVox (Fang et al., 2022) | 21.3 | 0.31 | 28.2 | 0.79 | 28.6 | 0.96 | 24.4 | 0.64 | **22.8** | **0.73** | 25.06 | 0.69 |
| 4DGS Wu et al. (2024) | 21.5 | 0.35 | 26.8 | 0.80 | 29.7 | 0.76 | 27.8 | 0.84 | 22.0 | 0.71 | 25.56 | 0.69 |
| +STDR | **22.4** | **0.38** | **28.9** | **0.86** | 30.3 | 0.78 | **28.0** | **0.86** | 22.2 | 0.72 | **26.36** | **0.72** |

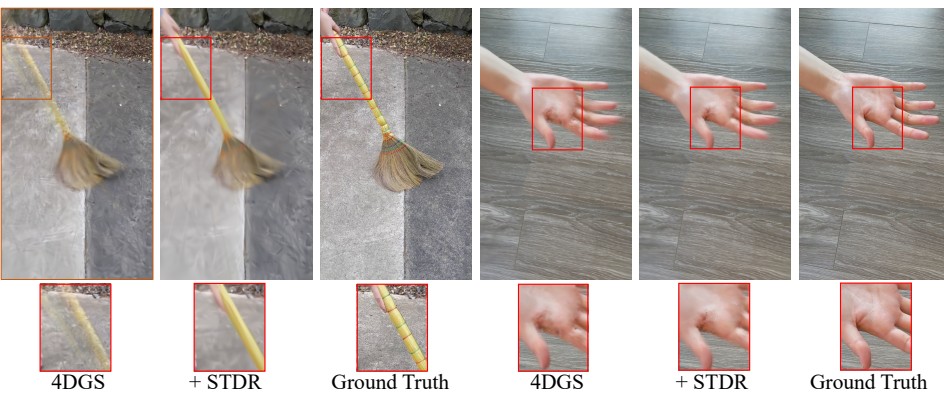

| 4DGS | + STDR | Ground Truth | 4DGS | + STDR | Ground Truth |

Figure 5: Visualization of Comparisons on HyperNeRF Dataset (Park et al., 2021b)

of motion patterns and topological changes, allowing a comprehensive assessment of reconstruction quality.

## A.2 MORE RESULTS

We integrate the proposed STDR module into 4DGS and conduct qualitative evaluations on the HyperNeRF dataset. As shown in Table 4, the reconstruction quality is significantly improved after incorporating our module. In addition, we provide visual analysis in Figure 5, where clear improvements can be observed. Specifically, in the broom scene, 4DGS exhibits noticeable spatio-temporal ghosting artifacts, which are effectively mitigated by our method, demonstrating the capability of STDR in enhancing temporal consistency. Meanwhile, we also present experiments on FPS with respect to the number of 3D Gaussians, conducted on an NVIDIA RTX 3090 GPU. As shown in Table 5, the results demonstrate that our method preserves real-time rendering performance even after integration, indicating its practical applicability in resource-constrained settings. We further visualize the canonical space reconstructed on the Lego scene from the D-NeRF dataset, as shown in Figure 6. The canonical space corresponds to the static Gaussian representation constructed during the initialization stage. In subsequent training, dynamic motion is modeled by deforming and activating these canonical Gaussians over time rather than reconstructing geometry from scratch. Therefore, the canonical spaces obtained in 4DGS, Deformable3D, and our method are largely consistent, as all are derived from the same static initialization process. Additionally, we integrate our method into DeformableGS and render depth maps on scenes from the D-NeRF and NeRF-DS datasets, demonstrating that it can accurately reconstruct the underlying geometric structure, as shown in Figure 7.

## A.3 ANALYSES AND DISCUSSIONS

**Q1: Why is opacity modulated using a spatio-temporal mask?**

**A1:** We modulate the opacity of each Gaussian using a learnable spatio-temporal mask to explicitly model temporal activation patterns and disentangle spatio-temporal relationships during the early stages of training. This design introduces several benefits over directly optimizing the original opacity values.

Table 5: Frame rates (FPS) and Gaussian counts for each scene across D-NeRF datasets.

| DeformGS | | | +STDR | | |
|---|---|---|---|---|---|
| Scene | FPS | Num (k) | Scene | FPS | Num (k) |
| Jump | 85 | 40 | Jump | 51 | 37 |
| Bouncing | 58 | 81 | Bouncing | 46 | 86 |
| T-Rex | 48 | 110 | T-Rex | 31 | 107 |
| Mutant | 46 | 94 | Mutant | 37 | 84 |
| Warrior | 135 | 20 | Warrior | 107 | 18 |
| Standup | 100 | 37 | Standup | 90 | 35 |
| Hook | 64 | 76 | Hook | 53 | 66 |

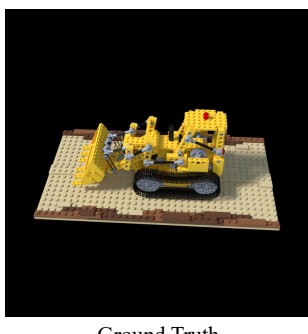 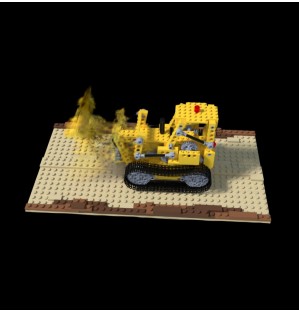 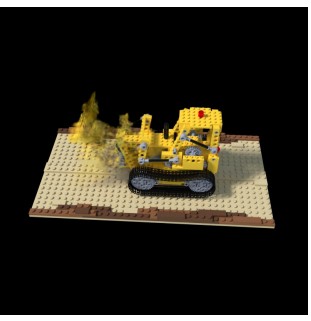

Ground Truth             Deform GS             +STDR

Figure 6: Visualization of Canonical Representation

First, the spatio-temporal mask provides a soft, probabilistic encoding of each Gaussian's temporal identity by assigning activation weights across timestamps. This allows the model to gradually associate each Gaussian with a specific temporal state, improving both the interpretability and temporal consistency of the representation. After a warm-up phase, the mask is normalized via a softmax function, yielding a spatio-temporal probability distribution that can be directly used for downstream modules such as temporal feature extraction.

Second, we freeze the original opacity $\alpha$ during the warm-up stage and restrict gradient flow to the spatio-temporal mask. This choice stabilizes training by preventing ambiguous gradients from modifying scene representations too early. Since canonical Gaussians are initialized from temporally entangled observations, directly optimizing $\alpha$ risks reinforcing ghosting artifacts and spatial overlaps. By modulating opacity through the mask, the model can first resolve temporal ambiguity before updating core Gaussian attributes.

Finally, the mask structure naturally reveals motion patterns. Gaussians in static regions tend to have uniform activation across time, while dynamic ones exhibit sharp, time-specific activations. This emergent property not only supports temporal reasoning but can also serve as auxiliary supervision for dynamic/static decomposition and motion-aware tasks.

**Q2: Why is spatio-temporal consistency regularization applied only during the warm-up and early training stages?**

**A2:** We apply spatio-temporal consistency regularization only during the warm-up phase and the early training iterations that follow. This is because the primary role of this regularization is to stabilize the learning of temporal activation patterns and promote local coherence when the spatio-temporal masks are still evolving. If the regularization is applied for too long, it may impose overly strong constraints on the temporal behavior of Gaussians, preventing the masks from naturally forming a probabilistic distribution that reflects the true dynamics of the scene.

**Q3: Why is a separated deformation field introduced in the proposed framework?**

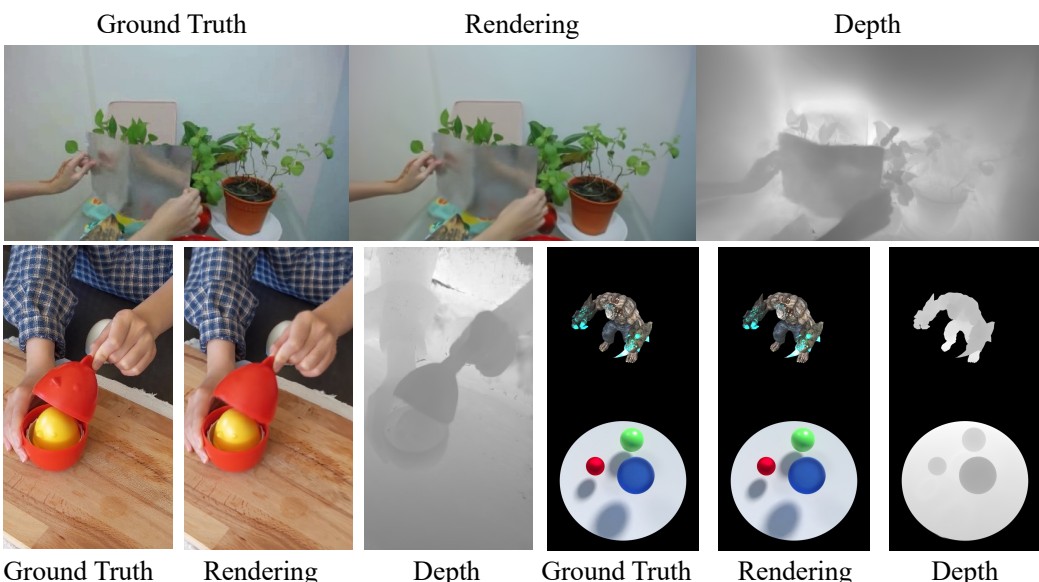

Figure 7: Rendering and Depth Visualization.

**A3:** The separated deformation field is specifically designed to disentangle spatial structure from temporal dynamics when modeling motion in dynamic scenes. Instead of relying on a single deformation field that entangles both spatial and temporal cues, we explicitly factor the learned spatio-temporal probability distribution of each Gaussian into two components: a spatial embedding and a temporal embedding. This separation allows the network to model static geometry and dynamic behavior with greater clarity and reduced interference between the two.

From a modular design perspective, this separation improves interpretability and composability. By isolating motion-specific features from geometry-aware features, each component can be optimized more effectively for its target objective—spatial alignment or temporal alignment—leading to more stable training and better generalization. Additionally, it provides a clean interface for integrating downstream tasks.

**Q4: Why is KL divergence used to compute the loss in the spatial-awareness regularization?**

**A4:** Compared to other loss functions such as L2 distance or cosine similarity, KL divergence offers a more informative and flexible way to align temporal distributions. While L2 and cosine metrics evaluate only pointwise similarity or directional alignment between two vectors, KL divergence considers the full structure of the probability distributions, including differences in scale, sparsity, and overall shape.

This property is particularly valuable for spatio-temporal regularization. In dynamic regions, neighboring Gaussians may activate at similar but not identical timestamps. KL divergence softly penalizes discrepancies without enforcing strict uniformity, allowing the model to maintain natural temporal variation while still promoting local coherence. As a result, it supports smoother deformation learning and enhances the stability and accuracy of dynamic scene reconstruction.

### A.4 LIMITATION

While STDR offers notable improvements in spatio-temporal consistency, it currently relies on a fixed temporal resolution determined by the input timestamps. This design works well under regular motion and balanced temporal sampling, but may face challenges when applied to scenes with highly non-uniform dynamics or missing observations at specific time steps. In such cases, the learned temporal distributions might be less expressive or over-smoothed. Nonetheless, this limitation primarily affects extreme scenarios, and STDR remains robust across a wide range of realistic

settings. Future extensions may consider adaptive temporal modeling strategies to further enhance generalization under diverse motion patterns.

## A.5   LLM USAGE

In this work, large language models (LLMs) were only used as a general-purpose writing assist tool. Specifically, LLMs were employed for correcting grammatical errors and refining the language style of the manuscript. No part of the research ideation, methodology design, experiments, analysis, or results was generated by LLMs. The authors take full responsibility for the content of this paper.

