# OpenReview forum: "STDR: Spatio-Temporal Decoupling for Real-Time Dynamic Scene Rendering"
_ICLR.cc/2026/Conference — Submitted to ICLR 2026_

### Official Review · Reviewer_dmef · 2025-10-26

**Soundness:** 4
**Presentation:** 4
**Contribution:** 3
**Rating:** 6
**Confidence:** 4

**Summary:**

This paper tackles the `spatio-temporal incoherence` that arises when dynamic scenes are initialized as if all frames share one timestamp, yielding ghosted, temporally mixed Gaussians that confuse downstream deformation learning. The authors propose **STDR**, a plug-and-play spatio-temporal decoupling module built for 3D Gaussian pipelines: (i) a learnable spatio-temporal mask that evolves into a probability distribution over timestamps, (ii) a Separated Deformation Field that factorizes spatial structure and temporal dynamics using those distributions, and (iii) a warm-start consistency regularization to stabilize training and encourage smooth, coherent motion. Experiments on dynamic-scene benchmarks report consistent reconstruction gains (with reasonable FPS) and demonstrate that the learned masks are interpretable (uniform for static regions, sharp activations for dynamic ones).

**Strengths:**

- This paper is well-written and easy to understand.
- Introducing the Spatio-Temporal Mask is particularly elegant. It simultaneously characterizes the Gaussians’ persistence and captures spatial smoothness, a very sensible design choice.
- By exploiting the mask’s spatial and temporal cues, the Separated Deformation Field offers a neat decoupling that, in theory, enables the network to learn a cleaner flow.
- The empirical results are compelling. Owing to its plug-and-play design, the method is compatible with essentially all deformation-field–based frameworks. Quantitative evaluations exhibit consistent improvements, and the visualizations align with these improvements.

**Weaknesses:**

- Missing references for some important dynamic reconstruction works:
  - [CVPR 2024] Spacetime Gaussian Feature Splatting for Real-Time Dynamic View Synthesis, by Zhan Li et al.
  - [CVPR 2025] FreeTimeGS: Free Gaussian Primitives at Anytime and Anywhere for Dynamic Scene Reconstruction, by Yifan Wang et al.
- Missing comparison on FPS and primitive counts. It would help to include a direct comparison of the number of Gaussian primitives and the corresponding FPS to clarify the runtime trade-offs.
- Given the central role of the spatio-temporal mask, I strongly recommend adding visualizations to demonstrate effectiveness. Specifically, to show that the learned mask aligns with static vs. dynamic regions and reflects spatial smoothness.
- **Fixed temporal resolution.** The method relies on a $K$-dimensional time distribution per Gaussian, which may over-smooth or underfit scenes with highly non-uniform motion or missing frames. The authors could try to explore adaptive/continuous-time parameterizations and report sensitivity to $K$.
- The paper would benefit from a direct comparison between the proposed mask mechanism and FreetimeGS’s opacity-persistence scheme. It is an instructive contrast in my view, with the mask offering a notably more elegant formulation.
- Minor typo errors:
  - L144-145: `is the color and the blending weight ` -> `are the color and the blending weight`
  - L191-192: `Gaussianss` -> `Gaussians`
  - L231-232: `sptio-temporal mask` -> `spatio-temporal mask`
  - Figure 2: `Orginial Deformation Field` -> `Original Deformation Field`
  - L321-315: `spation-temporal mask` -> `spatio-temporal mask`
  - L136: `render the influence of xxx` -> ` renders the influence of`
  - L498-499: `Section 4and Appendix A.` -> Add space between `4` and `and`

**Questions:**

1. What exact values do you use for $M$ and $K$? Are they fixed across all datasets/scenes, or tuned per dataset? It can be much better if the authors can further provide a **sensitivity analysis** for $M$ and $K$
2. Why is **SP-GS** used as the baseline on NeRF-DS instead of **Deformable-GS** and **SC-GS**, which appear elsewhere in the paper? I guess the choice is motivated by performance/training-time/FPS considerations?

**Details Of Ethics Concerns:**

None.

---

> ### Author Response · Authors · 2025-11-22
> **Rebuttal by Authors**
>
> We sincerely thank the reviewer for their positive and detailed feedback, especially the recognition of our spatio-temporal mask design, the separated deformation field, and the clarity of the paper. We also appreciate the acknowledgement of the plug-and-play nature and consistent empirical improvements, which strongly encourages us about the practicality and impact of our approach. In addition, we thank the reviewer for pointing out formatting issues and suggestions on related work; we have corrected these and incorporated the corresponding discussions in the revised manuscript.
>
> **Q1: Rendering Efficiency Comparison**
>
> While STDR introduces a modest increase in training time, we emphasize that **inference speed remains unaffected**, and the system continues to support **real-time rendering performance**, as detailed in Table 5 in the appendix of the main paper. This further underscores the practicality of STDR in dynamic 3DGS applications, even in **latency-sensitive scenarios**.
>
> **Q2: Visualization of Dynamic and Static Gaussians Across Timestamps**
>
> We further produce an animated visualization covering all timestamps, which clearly distinguishes dynamic Gaussians from static Gaussians and illustrates how different Gaussians evolve over time throughout the dynamic reconstruction process. (video: pointcloud_time-lego.gif, available at [https://anonymous.4open.science/r/stdr-885C/](https://anonymous.4open.science/r/stdr-885C/))
>
> **Q3: Sparse or Irregular Timestamp Scenarios**
>
> We have conducted additional experiments on the D-NeRF dataset by artificially downsampling or removing specific frames to simulate **sparse and irregular timestamp** conditions. As shown in **Table 1**, although baseline performance degrades due to reduced temporal information, integrating **STDR** consistently improves PSNR and SSIM, demonstrating strong robustness under irregular temporal sampling. This improvement arises because our spatio-temporal decoupling mechanism adjusts Gaussian opacity and deformation behavior based on the learned temporal prior, enabling STDR to adapt to variable frame rates, preserve relative motion patterns, and reduce temporal inconsistency.
>
> In our current implementation, the temporal resolution \(K\) for each Gaussian is chosen in relation to the number of training timestamps, so that each temporal component can leverage the available input frames to modulate opacity during training. This design keeps the model simple and stable across different datasets, while already providing clear improvements over baseline methods. We appreciate the reviewer’s suggestion, and we view more adaptive or continuous-time parameterizations of the temporal distribution as a promising direction for future work, which may further enhance flexibility in scenarios with highly non-uniform motion or severely undersampled temporal observations.
>
> **Table 1. Sparse-timestamp evaluation on the D-NeRF dataset.**
>
> | Method     | PSNR ↑ | SSIM ↑ | LPIPS ↓ |
> |------------|:------:|:------:|:--------:|
> | DeformGS   | 39.35  | 0.987  | 0.013    |
> | + STDR     | 40.20 | 0.991 | 0.011 |
> | SCGS       | 40.82  | 0.990  | 0.011    |
> | + STDR     | **41.56** | **0.994** | **0.008** |
>
> **Q4: Comparison with FreeTimeGS Opacity-Persistence Mechanism**
>
> We thank the reviewer for highlighting the connection to **FreeTimeGS** and for the positive comments on our mask design. While both methods modulate the temporal contribution of Gaussians, they do so in fundamentally different ways.
>
> **FreeTimeGS** builds a fully **4D Gaussian representation** in \((x, y, z, t)\). Each primitive carries its own motion function and a **temporal opacity (opacity-persistence) function**, which directly controls its lifespan along the time axis. This mechanism is tightly coupled with a specific 4D parameterization and regularization scheme, and is designed as part of a dedicated FreeTimeGS backbone for highly flexible dynamic modeling.
>
> By contrast, our method introduces a **discrete, probabilistic spatio-temporal mask in the canonical space**. This mask is
> (i) learned once during the **static initialization stage** and then **frozen as a temporal prior**,
> (ii) used to guide subsequent deformation or motion-field optimization, and
> (iii) implemented as a **lightweight, plug-and-play module** that can be attached to various canonical+deformation pipelines (DeformGS, SC-GS, SPGS, 4DGS, etc.) without changing their core 4D formulation.
> Critically, our canonical-stage prior provides something that FreeTimeGS does not explicitly model: a **structured spatio-temporal prior over Gaussians** that is shared across architectures and repeatedly reused throughout training. This prior explicitly captures how each Gaussian behaves over space and time, and can be plugged into different dynamic 3DGS backbones as a common guiding signal, instead of being tied to one specific representation design.

---

> > ### Author Response · Authors · 2025-11-22
> > **Rebuttal by Authors**
> >
> > **Q5: Sensitivity to M and K**
> >
> > We set M = 1000 and K = 5 for all datasets and scenes, without any per-dataset tuning. To further address the reviewer’s concern about sensitivity, we additionally conduct a small-scale sensitivity analysis on the D-NeRF dataset (DeformGS + STDR).
> >
> > For M, we vary M ∈ {500, 1000, 1500} while keeping K = 5 and all other settings fixed. As shown in Table 2, increasing M from 500 to 1000 brings a slight improvement, whereas further increasing M to 1500 yields only marginal gains. For K, we vary K ∈ {3, 5, 7} with M = 1000. For the temporal resolution K, we vary K ∈ {3, 5, 7} while fixing M = 1000. The results in Table 3 show that K = 5 achieves the best overall performance, whereas both smaller (K = 3) and larger (K = 7) values lead to slightly worse PSNR/SSIM and higher LPIPS.
> > **Table 2. Sensitivity to M on the D-NeRF dataset (DeformGS + STDR, K = 5).**
> >
> > | M      | PSNR ↑ | SSIM ↑ | LPIPS ↓ |
> > |--------|:------:|:------:|:-------:|
> > | 500    | 40.82  | 0.992  | 0.011   |
> > | 1000   | 41.00  | 0.993  | 0.010   |
> > | 1500   | 41.03  | 0.993  | 0.010   |
> >
> > **Table 3. Sensitivity to K on the D-NeRF dataset (DeformGS + STDR, M = 1000).**
> >
> > | K      | PSNR ↑ | SSIM ↑ | LPIPS ↓ |
> > |--------|:------:|:------:|:-------:|
> > | 3      | 40.88  | 0.992  | 0.011   |
> > | 5      | 41.00  | 0.993  | 0.010   |
> > | 7      | 40.92  | 0.992  | 0.011   |
> >
> > These results indicate that our method is **not overly sensitive** to the exact choices of M and K. We therefore use the shared default configuration (M = 1000, K = 5) across all experiments as a reasonable trade-off between performance and computational cost.
> >
> > **Q6: Rationale for Using SP-GS as the Baseline on NeRF-DS**
> >
> > We appreciate the reviewer’s question. Our initial intention was to evaluate STDR on diverse baselines across different datasets to demonstrate its general applicability rather than relying on a single architecture. On NeRF-DS, we chose SP-GS because it offers a good balance between reconstruction quality, training cost, and real-time performance. Following the reviewer’s suggestion, we additionally evaluated Deformable-GS on NeRF-DS, and observed a consistent improvement with STDR: Deformable-GS achieves 24.11 / 0.8525 / 0.1769 (PSNR / SSIM / LPIPS), while Deformable-GS + STDR achieves 24.48 / 0.8601 / 0.1755, further confirming the effectiveness and generality of our method.
> >
> > **Table 4. NeRF-DS results with Deformable-GS backbone (PSNR / SSIM / LPIPS).**
> >
> > | Method              | PSNR ↑ | SSIM ↑  | LPIPS ↓  |
> > |---------------------|:------:|:-------:|:--------:|
> > | Deformable-GS       | 24.11  | 0.8525  | 0.1769   |
> > | + STDR| **24.48** | **0.8601** | **0.1755** |

---

### Official Review · Reviewer_jdhU · 2025-10-27

**Soundness:** 3
**Presentation:** 3
**Contribution:** 2
**Rating:** 4
**Confidence:** 4

**Summary:**

This paper extends deformable 3D Gaussian Splatting (3DGS) by introducing an additional temporal opacity mechanism to improve temporal modeling in dynamic scene reconstruction. The proposed STDR module combines a spatio-temporal mask, separated deformation field, and consistency regularization to explicitly decouple spatial and temporal components during training. The method achieves better quantitative and qualitative results than several baselines, including SP-GS, SC-GS, and DeformGS.

**Strengths:**

1. Performance Gains: The method improves reconstruction quality across both synthetic and real-world datasets, outperforming reported baselines such as SP-GS and DeformGS.

2. Integration: The module is **plug-and-play** and compatible with multiple existing 3DGS pipelines.

**Weaknesses:**

1. Inaccurate or Overgeneralized Claims

   * The statement *“existing 3DGS-based methods typically adopt a two-stage pipeline”* is not universally true. Several recent dynamic Gaussian methods (e.g., 4DGS,) employ different initialization strategies.
   * The claim that the proposed masks *“reflect the true dynamics of the scene”* is too strong — if the temporal sampling rate of training data is limited, this convergence cannot guarantee ground-truth dynamic fidelity.

   * The assertion "first to introduce a spatio-temporal mask" may be overstated. The proposed mask effectively functions as a temporal modulation of Gaussian opacity, conceptually similar to prior opacity weighting or temporal blending in 4DGS's varaints.



2. Dataset Limitations and Overinterpretation:

   * The synthetic scenes used (e.g., D-NeRF) consist of simple deformations or rigid motions defined in Blender, which are easily modeled by MLPs. Improvements on these datasets do not necessarily validate claims of “true dynamics modeling,”.

**Questions:**

1. Real-world Extrapolation results:

   Are real-world evaluations are performed on interpolated camera trajectories? if so, please include extrapolation tests to validate that the model generalizes beyond the temporal training window if it claims to capture true scene dynamics.

2. Runtime & Scalability Comparisons:
   Report the training and inference time compared to SP-GS and DeformGS.
   Clarify whether scaling up existing baselines (e.g., higher-capacity deformation fields) could achieve similar quality without additional temporal masking.

3. Real-time Demonstration:
   If available, demonstrate integration into a GUI or visualization tool, rather than offline inference.

---

> ### Author Response · Authors · 2025-11-22
> **Rebuttal by Authors**
>
> We sincerely thank the reviewer for acknowledging our performance improvements, and for recognizing the plug-and-play compatibility of our module with existing 3DGS pipelines.
>
> **Q1: Clarification on the Scope of Real-World Dynamic Reconstruction**
>
> We thank the reviewer for this thoughtful question. We would like to clarify that the goal of dynamic 3D Gaussian reconstruction is not to predict future motion beyond the temporal training window, but rather to faithfully reconstruct the dynamic scene within the observed time range, especially under sparse-view and temporally varying inputs. The core challenge lies in recovering consistent geometry and motion trajectories by aggregating multi-view information across time, enabling the model to reconstruct regions that are missing, partially occluded, or sparsely observed at certain timestamps. Our method is designed specifically to address this challenge by introducing spatio-temporal regularization in the canonical representation, thereby improving temporal consistency and robustness under real-world view sparsity.
>
> Moreover, we note that performing true temporal extrapolation is intrinsically challenging under the current setting. If no images are available at a future timestamp, the model has no reliable observations to condition on, and any “reconstruction” would necessarily rely on uncontrolled hallucination rather than verifiable geometry. In addition, existing dynamic-scene benchmarks only provide data within the captured time window, so there is no ground-truth supervision or standard evaluation protocol for frames beyond this range. Consequently, this limitation arises from the nature of the available datasets and task definition, rather than from a deliberate omission of such experiments on our side.
>
> **Q2: Reconstruction and Rendering Efficiency Comparison**
>
> We further evaluate the overhead introduced by STDR by measuring **training time** and **peak GPU memory usage** on the D-NeRF and NeRF-DS dataset when integrating STDR into several dynamic 3DGS backbones. The results are summarized in Tables 1 and 2. As shown in **Table 1**, integrating STDR into DeformGS and SCGS leads to only moderate increases in training time (around **15–20%**) and peak memory (on the order of **several hundred MB to about 1 GB**). On the stronger SPGS backbone (**Table 2**), training time increases from 160 to 174 minutes (≈**8.8%**), and peak memory increases from 7.68 GB to 9.78 GB.
>
> **Table 1. Average training time (minutes) and peak GPU memory usage (MB) on the D-NeRF dataset (DeformGS / SCGS backbones).**
>
> | Method    | Time ↓ (min) | Mem ↓ (MB) |
> |:----------|:------------:|:----------:|
> | DeformGS  | **38**           | 2395       |
> | + STDR    | 48       | 3165   |
> | SCGS      | 43           | **1362**       |
> | + STDR    | 58       | 1694   |
>
> **Table 2. Average training time (minutes) and peak GPU memory usage (MB) on the NeRF-DS dataset (SPGS backbone).**
>
> | Method | Time ↓ (min) | Mem ↓ (MB) |
> |:-------|:------------:|:----------:|
> | SPGS   | **160**          | **7679.5**     |
> | + STDR | 174      | 9779.3 |
>
> Furthermore, while STDR introduces a modest increase in training time, we emphasize that **inference speed remains unaffected**, and the system continues to support **real-time rendering performance**, as detailed in **Table 5** in the Appendix of the main paper. This further underscores the practicality of STDR in dynamic 3DGS applications.
>
>
> **Q3: Clarification on Real-Time Demonstration**
>
> We thank the reviewer for the suggestion. Our results are not limited to offline inference. We have already integrated our system into a lightweight **GUI-based visualization tool**, which allows real-time interaction with the reconstructed dynamic scene. Users can freely explore the scene at **arbitrary viewpoints, positions, and timestamps**, demonstrating that our method supports real-time rendering and practical usage beyond static offline evaluation. We further provide a [real-time interactive visualization **video**: visual.gif (https://anonymous.4open.science/r/stdr-885C/)]( https://anonymous.4open.science/r/stdr-885C/) to substantiate this claim.

---

> > ### Author Response · Authors · 2025-11-22
> > **Rebuttal by Authors**
> >
> > **Q4: Clarification on Applicability Scope and Dataset Coverage**
> >
> > We thank the reviewer for pointing out these issues and for highlighting places where our descriptions can be made more precise. We agree that the statement “existing 3DGS-based methods typically adopt a two-stage pipeline” does not strictly apply to all recent dynamic Gaussian methods. We will revise the text to explicitly state that our method is applicable to dynamic reconstruction frameworks that follow a **two-stage pipeline** and satisfy two key conditions:
> >
> > (1) a **static initialization stage**, where we introduce the spatio-temporal probability distribution to capture the decoupled spatial–temporal behavior of Gaussians; and
> >
> > (2) a **subsequent deformation or motion modeling stage**, in which the probability prior learned in the first stage provides effective guidance for temporal optimization.
> >
> > Representative dynamic Gaussian-based methods that meet these criteria include **Deformable3DGS**, **SC-GS**, **SPGS**, **4DGS**, **HAIF-GS**, **Grid-4D**, **MotionGS**, and many other related methods that perform canonical static reconstruction followed by temporal deformation or motion-field learning. Other dynamic rendering approaches (e.g., **Dynamic3DGS**) may also integrate our module with minor adaptations. In the revised version, we will clarify this applicability scope more explicitly and avoid wording that might suggest universal coverage.
> >
> > We also agree that synthetic datasets such as **D-NeRF** have limited ability to fully reflect real-world dynamics. However, these benchmarks are standard and widely used to evaluate dynamic 3D representations in a controlled setting. To address this concern, our evaluation is **not limited to synthetic data**: we additionally conduct quantitative and qualitative comparisons on real-capture datasets such as **NeRF-DS** and **HyperNeRF**, where camera motion, lighting, and object motion are more complex and noisy. Across both synthetic and real-world datasets, our method consistently improves temporal consistency and geometric stability over strong baselines. We will make this point clearer in the revised manuscript.
> >
> > **Q5: Clarification on Capacity of the Deformation Field**
> >
> > We conducted an additional experiment where we only increase the capacity of the deformation field in DeformGS, without using our spatio-temporal mask. Concretely, we double the depth of the deformation MLP from 8 layers to 16 layers (“DeformGS-Large”), while keeping all other settings unchanged. As summarized in **Table 3**, the high-capacity variant brings even degrades reconstruction quality, while significantly increasing the number of deformation parameters and per-iteration runtime.
> >
> >
> > These results suggest that blindly increasing deformation capacity is insufficient to resolve temporal inconsistency and geometric instability. Our probabilistic spatio-temporal prior provides structured temporal guidance, which cannot be replicated by simply stacking more layers in the deformation MLP.
> >
> > **Table 3. Comparison between increasing deformation capacity and adding STDR (Hook Scene of D-NeRF dataset).**
> >
> > | Method             | PSNR ↑ | SSIM ↑ | LPIPS ↓ |
> > |--------------------|:------:|:------:|:-------:|
> > | DeformGS (8)       | 37.42  | 0.987  | 0.014   |
> > | DeformGS-Medium (10) | 35.99  | 0.983  | 0.017   |
> > | DeformGS-Large (16)  | 32.16  | 0.965  | 0.039   |
> > | DeformGS + STDR    | **38.17** | **0.989** | **0.012** |

---

> > > ### Comment · Reviewer_jdhU · 2025-11-28
> > >
> > > I’m satisfied with the rebuttal, but I’m currently unable to revise the score.

---

> > > > ### Author Response · Authors · 2025-11-28
> > > >
> > > > We would like to sincerely thank Reviewer jdhU for the thoughtful review and constructive suggestions, which have helped us improve the paper. We are deeply grateful for the time and effort you devoted to carefully evaluating our work, and we truly appreciate your helpful guidance and encouragement.

---

### Official Review · Reviewer_7iiN · 2025-10-31

**Soundness:** 2
**Presentation:** 2
**Contribution:** 2
**Rating:** 4
**Confidence:** 3

**Summary:**

This paper presents a plug-and-play module designed to enhance dynamic scene reconstruction in 3D Gaussian Splatting. It addresses the issue of spatio-temporal entanglement by learning separate spatial and temporal probability distributions for each Gaussian. Through a spatio-temporal mask, a decoupled deformation field, and consistency regularization, STDR effectively disentangles motion and structure.

**Strengths:**

1. I think decoupling spatial-temporal modeling makes sense, though I think this lacks novelty given that extensive approaches address this issue [A]. I think it is easy to find massive approaches for decoupling spatial-temporal modeling.
2. The proposed approach is plug-and-play, and can be applied to Deformable3DGS, SC-GS, SPGS on separated benchmarks.

[A] SDD-4DGS: Static-Dynamic Aware Decoupling in Gaussian Splatting for 4D Scene Reconstruction

**Weaknesses:**

1. My first concern is that this paper only presents a few approaches for plug-and-play evaluation. Meanwhile, those baselines are not representative enough for different lines of work. I think this approach is not appliable for all previous appoaches, could u explain which  kinds of paper are suitable for STDR?
2. In 4DGS, some work illustrates the plug-and-play attribute. Could u discuss the differences? [C]
3. It is confusing that this does not apply the proposed technique to Spatial-Temporal Decoupling approaches, e.g. [D]

[C] TimeFormer: Capturing Temporal Relationships of Deformable 3D Gaussians for Robust Reconstruction
[D] Hybridgs: Decoupling transients and statics with 2d and 3d Gaussian splatting

**Questions:**

Overall, my major concern is the novelty and the evaluation of effectiveness.

---

> ### Author Response · Authors · 2025-11-22
> **Rebuttal by Authors**
>
> We sincerely thank the reviewer for their constructive comments and valuable insights. We appreciate their recognition of the soundness of our spatio-temporal decoupling design and the convenience of its plug-and-play integration.
>
> **Q1: Clarification on the Scope of Plug-and-Play Applicability**
>
> Our method is applicable to dynamic reconstruction frameworks that follow a **two-stage pipeline**, satisfying two key conditions:
>
> (1) a **static initialization stage**, where our spatio-temporal probability distribution is introduced to capture the decoupled spatio–temporal relationship of Gaussians; and
> (2) a **subsequent deformation or motion modeling stage**, where the learned probability prior from the first stage provides effective guidance for temporal optimization.
>
> Representative dynamic Gaussian-based methods that meet these criteria include **Deformable3DGS**, **SC-GS**, **SPGS**, **4DGS**, **HAIF-GS**, **Grid-4D**, **MotionGS**, **and many other related methods** that also adopt canonical static reconstruction followed by temporal deformation or motion-field learning.
>
> Some other dynamic rendering methods (e.g., Dynamic3DGS) may also integrate our module with minor adaptation. We acknowledge that our method is not intended to cover all existing 4D modeling paradigms. Instead, our goal is to propose a **lightweight and general enhancement** that can be incorporated into representative Gaussian Splatting frameworks to improve **temporal consistency** and **geometric stability**. We will clarify this applicability scope more explicitly in the revised version.
>
> **Q2: Clarification on the Difference from TimeFormer and Other Plug-and-Play Designs**
>
> We thank the reviewer for pointing out TimeFormer as a relevant plug-and-play design. We would like to emphasize that TimeFormer and our method operate at different levels of the dynamic 3DGS pipeline.
>
> TimeFormer improves dynamic 3DGS by adding an additional temporal encoder: it takes Gaussian states from multiple timestamps as input to a Transformer in order to model temporal relationships. In this way, it enhances temporal modeling by introducing a high-capacity, feature-level temporal module on top of an existing dynamic 3DGS backbone.
>
> In contrast, our method does not introduce an extra high-capacity temporal network. Instead, we modify the Gaussian representation itself by assigning each Gaussian a spatio-temporal probability distribution that is learned once during the static initialization stage. This distribution is then used as a temporal prior to guide the subsequent deformation stage, indicating when each Gaussian should be active and how it should participate in the learned motion.
>
> Moreover, our temporal prior is compatible with TimeFormer rather than competing with it. The learned spatio-temporal probabilities can be fed into TimeFormer as additional conditioning signals, providing explicit guidance on which regions are likely static or dynamic. In this sense, TimeFormer focuses on learning temporal correlations, whereas our method provides a structured temporal prior; the two are conceptually complementary and can be combined.
>
> **Q3: Clarification on the Applicability to Spatio–Temporal Decoupling Methods**
>
> We thank the reviewer for pointing out **HybridGS** as a relevant spatio–temporal decoupling method. However, HybridGS has not released its training code and only provides partial rendering utilities. Since our method is integrated at the representation level during the canonical initialization and deformation training stages, the absence of full training code makes a fair and complete experimental comparison infeasible.
>
> Importantly, our approach is conceptually non-conflicting with HybridGS. HybridGS addresses static–transient separation through a hybrid 2D–3D Gaussian formulation, whereas our method introduces a **probabilistic spatio-temporal prior** that regularizes temporal behavior at the representation level. These two ideas are orthogonal and could be complementary once full training support becomes available.
>
> As a substitute for a direct comparison with HybridGS, we provide an additional experiment against another spatio–temporal decoupling–style baseline, **GaussianPrediction**, on the D-NeRF dataset. As shown in Table 1, integrating STDR into DeformGS achieves higher PSNR and SSIM and lower LPIPS than both the original DeformGS and GaussianPrediction, indicating that our spatio-temporal prior brings consistent gains over a representative decoupling-based baseline.
>
> **Table 1. Additional comparative experiments on the D-NeRF dataset.**
>
> | Method             | PSNR ↑ | SSIM ↑ | LPIPS ↓ |
> |--------------------|:------:|:------:|:-------:|
> | GaussianPrediction | 40.58  | 0.992  | 0.012   |
> | DeformGS           | 40.43  | 0.992  | 0.011   |
> | DeformGS + STDR    | **41.00**  | **0.993**  | **0.010**   |

---

### Official Review · Reviewer_NPP7 · 2025-11-02

**Soundness:** 2
**Presentation:** 3
**Contribution:** 2
**Rating:** 4
**Confidence:** 3

**Summary:**

This paper proposes STDR, a method to resolve the spatio-temporal incoherence that arises during canonical Gaussian initialization, when multi-frame observations are aggregated without temporal distinction. Specifically, under the per-scene optimization framework, STDR first assigns for each Gaussian a vector of temporal activation probabilities. The deformation field is modeled as a factorized deformation field that considers both spatial and temporal features for disentangled motion learning. Spatio-temporal consistency regularization is added on top to improve spatial and temporal smoothness. The paper conducts extensive experiments on synthetic (D-NeRF) and real-world (NeRF-DS, HyperNeRF) datasets, demonstrating consistent improvements in PSNR, SSIM, and LPIPS. The method is found to reduce ghosting artifacts and enhance temporal alignment.

**Strengths:**

- The paper is well-written and the proposed method is clearly explained.
- The proposed method is a plug-and-play module that can be easily integrated. When applied to different baseline methods (such as Deformable3D, SCGS, and SPGS), the method shows compatibility and effectiveness, consistently increasing the qualitative performance.

**Weaknesses:**

- The novelty of the proposed spatio-temporal probability distribution is somewhat limited. In related works such as 4DGS, the opacity of the Gaussians is also modulated by the temporal dimension, and the standard deviation reflects the persistence of the Gaussian. The proposed method seems to be a discretized version where a long vector whose length is proportional to the number of frames has to be introduced for every single Gaussian.
- Rendering efficiency is not reported. By introducing the spatio-temporal probability distribution, additional overhead is incurred by having to recompute the Gaussian opacity for each new timestamp. This would also lead to a potentially higher number of Gaussians, since the Gaussians could be simply turned opaque just at the timestamp where it gets observed, instead of building a canonical space and deforming over it. It would be interesting to visualize the 4D tracks of subsets of Gaussians.
- There are several typos in the paper. For example, in Eq.(1), the definition of the 3DGS should have $X-\mathcal{X}$ instead of just $\mathcal{X}$. Another example is on L231, "sptio" -> "spatio".

**Questions:**

- It would be interesting to demonstrate the canonical space built by the model. If part of the geometry is not visible from one specific timestamp, will that part be visible in the final reconstruction by warping from other frames?
- What is the reconstruction and final rendering efficiency in comparison to other baseline methods?
- Will the method be able to reconstruct proper geometry in the scene dataset? This could be better illustrated by visualizing the depth maps reconstructed from the NeRF-DS dataset or in an interactive viewer.

---

> ### Author Response · Authors · 2025-11-22
> **Rebuttal by Authors**
>
> We sincerely thank the reviewer for their encouraging evaluation. We appreciate the acknowledgment of our work’s clarity and the effectiveness of our plug-and-play design across diverse baselines. This positive feedback reinforces our confidence in the practicality and generality of the proposed framework.
>
> **Q1: Discussion on Canonical Representation and Temporal Aggregation**
>
> The canonical space corresponds to the static Gaussian representation constructed during the initialization stage. In subsequent training, dynamic motion is modeled by deforming and activating these canonical Gaussians over time rather than reconstructing geometry from scratch. Therefore, the canonical spaces obtained in 4DGS, DeformGS, and our method are largely consistent, as all are derived from the same static initialization process. The main difference lies in how the temporal evolution is modeled and regularized in the following stage. We further visualize the canonical space reconstructed on the Lego scene from the D-NeRF dataset, as shown in **Figure 6** in main paper’s Appendix.
>
> In datasets such as **D-NeRF** and **NeRF-DS**, objects move continuously while the camera simultaneously changes viewpoints, naturally leading to temporary occlusions. Regions that are invisible at a certain timestamp become visible at other times, which allows our canonical representation to recover these regions through temporal warping and feature aggregation across frames.
>
> For example, in the *Lego* scene, the excavator arm moves upward while the camera orbits from left to right. Although the left side of the excavator becomes occluded in later viewpoints, our method can still reconstruct the entire upward motion within the canonical space from that viewpoint. This demonstrates that our approach effectively leverages spatio-temporal correspondences across frames to recover the geometry of regions that are temporarily invisible. We further provide a [real-time interactive visualization **video**: visual.gif (https://anonymous.4open.science/r/stdr-885C/)]( https://anonymous.4open.science/r/stdr-885C/) to substantiate this claim.
>
> **Q2: Reconstruction and Rendering Efficiency Comparison**
>
> We have further supplemented our study with a comparative experiment evaluating the **training time** and **peak GPU memory usage** of STDR when integrated into DeformGS and SCGS on the D-NeRF dataset, as presented in **Table 1**. When integrated into DeformGS and SCGS, STDR introduces only minor increases in training time (approximately **15–20%**) and peak GPU memory (**less than 1 GB**), which are negligible given modern hardware. These results confirm that STDR enhances temporal consistency with minimal additional cost, demonstrating its practicality for real-world dynamic 3DGS applications.
>
> **Table 1. Average training time (minutes) and peak GPU memory usage (MB) on the D-NeRF dataset.**
>
> | Method  | Time ↓ (min) | Mem ↓ (MB) |
> |:--------|:------------:|:----------:|
> | DeformGS | **38** | 2395 |
> | + STDR   | 48 | 3165|
> | SCGS    | 43 | **1362** |
> | + STDR  | 58 | 1694 |
>
> Furthermore, while STDR introduces a modest increase in training time, we emphasize that inference speed remains unaffected, and the system continues to support **real-time rendering performance**, as detailed in **Table 5** in the Appendix of the main paper. This further underscores the practicality of STDR in dynamic 3DGS applications.
>
> **Q3: Geometric Reconstruction Quality on Scene Datasets and Visualization of Dynamic Gaussian Trajectories**
>
> We further supplement our study with depth reconstruction experiments. As illustrated by the **visualized depth maps** in **Figure 7** of the **main paper’s Appendix**, STDR accurately recovers fine-grained geometry even in regions that are partially or temporarily invisible. These visualizations confirm that our spatio-temporal decoupling preserves geometric consistency and enables high-fidelity 3D reconstruction in dynamic scenes.
>
> We further provide a **visualization of dynamic Gaussian point trajectories per timestamp** (**video**: *pointcloud_time-lego.gif*, available at [https://anonymous.4open.science/r/stdr-885C/](https://anonymous.4open.science/r/stdr-885C/)), which clearly illustrates how each Gaussian evolves over time and substantiates our claim about improved spatio-temporal consistency.

---

> > ### Author Response · Authors · 2025-11-22
> > **Rebuttal by Authors**
> >
> > **Q4: Discussion on Spatio-Temporal Probability Design**
> >
> > We first clarify that our method does not conflict with 4DGS; rather, it serves as a **complementary mechanism** that can be seamlessly integrated into its training pipeline without modifying or replacing the core framework of 4DGS.
> >
> > Compared with 4DGS, although 4DGS also modulates Gaussian opacity along the temporal dimension, it introduces time \(t\) as a continuous variable only after the static initialization stage, by embedding \(t\) into the Gaussian parameters to form a 4D representation \((x, y, z, t)\) and modeling opacity variation over time to capture dynamics. This process, however, does not fully exploit the spatio-temporal prior information implicit in the initialization stage. In contrast, we propose a **discrete, interpretable, and structured spatio-temporal prior** that 4DGS does not explicitly model. Concretely, during static initialization, 4DGS does not incorporate temporal information and therefore lacks the ability to distinguish which Gaussians are temporally stable and which correspond to dynamic regions, whereas our spatio-temporal probability distribution precisely fills this gap. By introducing a probabilistic temporal prior at the static stage, STDR can already separate stable and dynamic regions at the early phase of training and provide effective guidance for subsequent deformation optimization, leading to more reasonable allocation of representational capacity, stronger temporal consistency, and more stable convergence.

---

### Author Response · Authors · 2025-12-03
**Author Final Remarks by Authors**

We sincerely thank the area chair for overseeing the review process and all reviewers for their thorough and constructive feedback. We are encouraged by the recognition of our plug-and-play spatio–temporal decoupling design and its consistent improvements across multiple dynamic 3DGS backbones. Below, we summarize each reviewer’s main concerns, our rebuttal actions (including new experiments and clarifications), and the post-discussion status.

| Reviewer | Concerns | Our Rebuttal | Feedback |
|---------|----------|--------------|----------|
| **NPP7** | **A.** canonical representation  **B.** temporarily invisible/occluded regions  **C.** training-time/memory overhead/rendering efficiency **D.** geometric reconstruction/visualizations of dynamic trajectories **E.** ours vs 4DGS | **A–B.** canonical space clarification; Lego canonical (Appendix Fig. 6), and a real-time interactive visualization **video**: [visual.gif](https://anonymous.4open.science/r/stdr-885C/) for recovering partially invisible regions via cross-time warping and aggregation.  **C.** timing/memory analysis on D-NeRF (Table 1): ~15–20% extra training time, <1 GB extra peak memory on DeformGS/SCGS, unchanged real-time inference (Appendix Table 5) **D.** depth reconstructionsd (Appendix Fig. 7), and dynamic-trajectory visualizations: [pointcloud_time-lego.gif](https://anonymous.4open.science/r/stdr-885C/)） showing fine-grained geometry and stable temporal behavior **E.** complementary to 4DGS via a discrete, interpretable spatio–temporal prior at static initialization, separating stable vs dynamic Gaussians and guiding later deformation | Concerns addressed but reviewer could not provide further feedback due to system issues. |
| **7iiN** | **A.** scope of applicability **B.** conceptual difference from other temporal modules **C.** discussion of related methods | **A.** precisely defined the applicability scope **B.** clarified that TimeFormer adds a high-capacity temporal encoder on top of the backbone, whereas STDR modifies the Gaussian representation itself via a learned spatio–temporal probability prior at initialization **C.** HybridGS training code unavailable (no fair direct comparison); conceptually orthogonal (Hybrid 2D–3D separation vs probabilistic spatio–temporal prior); added D-NeRF comparison vs GaussianPrediction (Table 1), and clarifies scope and plug-and-play positioning. | Concerns addressed but reviewer could not provide further feedback due to system issues. |
| **jdhU** | **A.** Scope of real-world dynamic reconstruction **B.** training time/memory/rendering efficiency **C.** real-time demonstration **D.** applicability to broader dynamic pipelines and real vs synthetic datasets **E.** effect of simply increasing deformation-field capacity | **A.** clarified task focus: faithful reconstruction within the observed time window under sparse views/occlusions (not future prediction); extrapolation is ill-posed without future images or ground-truth **B.** timing/memory studies on D-NeRF & NeRF-DS (Tables 1–2): ≈8–20% extra training time, moderate memory increase, real-time inference preserved (Appendix Table 5) **C.** GUI-based real-time visualization tool with interactive exploration and demo video (*visual.gif*) **D.** two-stage canonical+deformation pipelines; evaluations on synthetic (D-NeRF) and real-capture datasets (NeRF-DS, HyperNeRF) **E.** deformation-capacity ablation (DeformGS-Medium/Large vs DeformGS + STDR, Table 3): larger MLP degrades quality and increases runtime, while STDR improves PSNR/SSIM and lowers LPIPS | Reviewer was satisfied with our rebuttal and expressed willingness to raise the score, but could not update it due to system limitations.|
| **dmef** | **A.** rendering efficiency **B.** visualization of static vs. dynamic Gaussians **C.** robustness under sparse or irregular timestamps **D.** relationship to FreeTimeGS **E.** sensitivity to hyperparameters \(M\) and \(K\).  **F.** choice of SP-GS as baseline | **A.** modest training-time increase only; inference speed and real-time performance unchanged (Appendix Table 5) **B.** animated all-timestamp visualization (*pointcloud_time-lego.gif*) highlighting static vs dynamic Gaussians and their evolution **C.** sparse/irregular timestamp experiments on D-NeRF (Table 1) **D.** compared with FreeTimeGS’s backbone-specific 4D opacity-persistence design, our method provides a lightweight canonical spatio-temporal prior that can be easily reused across different dynamic 3DGS architectures. **E.** sensitivity study on \(M\) and \(K\) (Tables 2–3): stable performance **F.** NeRF-DS results with Deformable-GS (Table 4): +STDR improves PSNR/SSIM/LPIPS, confirming generality beyond SP-GS | Concerns addressed but reviewer could not provide further feedback due to system issues.|

We sincerely thank the area chair and all reviewers again for their invaluable feedback and effort.

---

### Meta-Review · Area_Chair_5mke · 2026-01-04

**Summary:**

This paper introduces STDR, a plug-and-play spatio–temporal decoupling module for dynamic 3D Gaussian Splatting, intended to mitigate spatio–temporal incoherence caused by canonical initialization. The paper is clearly written, and reviewers acknowledge that integrating STDR into several existing dynamic 3DGS pipelines leads to consistent but relatively modest quantitative improvements. However, the overall contribution is limited by a lack of clear methodological novelty and insufficient experimental evidence to support the claimed advantages. Although the rebuttal provides additional analyses and results, these additions do not substantially alter the core assessment of the paper. Therefore, the submission does not meet the ICLR acceptance bar, and the recommended decision is reject.

**Reviewer Concerns:**

While the rebuttal successfully clarified several implementation details and addressed some empirical questions, the conceptual novelty of the proposed approach may still not be fully established. In particular, the spatio–temporal probability mask appears closely related to existing opacity or blending mechanisms used in prior methods, and the manuscript does not yet clearly articulate a distinct modeling perspective or theoretical justification that differentiates STDR from these approaches. Moreover, several analyses presented in the rebuttal, including additional ablations, efficiency evaluations, and qualitative visualizations of reduced spatio–temporal artifacts, are not incorporated into the main manuscript, making it difficult to assess robustness and generality based on the paper alone.

**Reviewer Scores:**

The paper initially received three borderline reject scores and one borderline accept. Although one reviewer indicated an increased score, based on the rebuttal it is likely that other reviewers’ core concerns are not fully resolved, and the scores are therefore expected to remain unchanged.

---

### Decision · Program_Chairs · 2026-01-26

Reject